# Effect of Sugar- and Polyphenol-Rich, Diluted Cloudy Apple Juice on the Intestinal Barrier after Moderate Endurance Exercise and in Ultra-Marathon Runners

**DOI:** 10.3390/nu16091353

**Published:** 2024-04-30

**Authors:** Sarah Valder, Raphaela Staltner, Daniel Alexander Bizjak, Tuba Esatbeyoglu, Volker Herdegen, Magdalena Köpsel, Tihomir Kostov, Ina Bergheim, Patrick Diel

**Affiliations:** 1Department of Molecular and Cellular Sports Medicine, German Sports University Cologne, 50933 Cologne, Germany; valder@heim-spiele.com (S.V.); t.kostov@dshs-koeln.de (T.K.); 2Department of Nutritional Sciences, Molecular Nutritional Science, University of Vienna, 1090 Vienna, Austria; raphaela.staltner@univie.ac.at (R.S.); ina.bergheim@univie.ac.at (I.B.); 3Division of Sports and Rehabilitation Medicine, University Hospital Ulm, 89075 Ulm, Germany; daniel.bizjak@uniklinik-ulm.de; 4Department of Molecular Food Chemistry and Food Development, Institute of Food and One Health, Gottfried Wilhelm Leibniz University Hannover, 30167 Hannover, Germany; esatbeyoglu@foh.uni-hannover.de (T.E.); koepsel@foh.uni-hannover.de (M.K.); 5Eckes-Granini Group GmbH, 55268 Nieder-Olm, Germany

**Keywords:** endotoxin, endurance exercise, health, intestinal barrier, apple juice

## Abstract

Background: Exercise and the consumption of sugars result in a dysfunction of the intestinal barrier (IB). Here, we determined the effect of sugar in a natural matrix on the intestinal barrier after moderate (A) and intensive endurance exercise (B). Method: The IB function was determined before (pre) and after running (post), and 120 and 180 min after consuming the drink by measuring serum endotoxin concentrations (lipopolysaccharides—LPS), IL-6, CD14, and i-FABP. In study A, nonspecifically trained participants (*n* = 24, males and females, age 26 ± 4) ran for one hour at 80% of their individual anaerobic threshold (IAT). After finishing, the runners consumed, in a crossover setup, either 500 mL of water, diluted cloudy apple juice (test drink), or an identical drink (placebo) without the fruit juice matrix (FJM). In study B, the participants (*n* = 30, males and females, age 50 ± 9) completed an ultra-marathon run, were divided into groups, and consumed one of the above-mentioned drinks. Results: Study A: Exercise resulted in a significant increase in serum LPS, i-FABP, and IL-6, which decreased fast after finishing. No impact of the different drinks on LPS i-FABP, or IL-6 could be observed, but there was an impact on CD14. Study B: The ultra-marathon resulted in a strong increase in serum LPS, which decreased fast after finishing in the water and test drink groups, but not in the placebo group. Conclusions: The consumed drinks did not affect the kinetics of IB regeneration after moderate exercise, but impacted CD14 serum concentrations, indicating possible beneficial effects of the FJM on the immune system. After an ultra-marathon, IB function regenerates very fast. The intake of sugar (placebo) seems to have had a negative impact on IB regeneration, which was diminished by the presence of the FJM.

## 1. Introduction

The consumption of drinks containing carbohydrates during or after exercise has been described as providing energy (during) and supporting recovery processes after exercise [1,2]. Therefore, such drinks are intensively advertised and their consumption has become very popular among athletes [3]. Unfortunately, both exercise and sugar consumption have been described as resulting in a dysfunction of the intestinal barrier (IB). Impairments of the intestinal barrier function can subsequently lead to elevated bacterial endotoxin levels and inflammation [4,5]. With respect to physical activity, the induction of inflammation is, on the one hand, important for the stimulation of exercise-related physiological adaptations [6,7]; on the other hand, chronic inflammation may result in overtraining scenarios [8] and increase the risk for the development of metabolic diseases (for an overview, see [9]). In several studies, it has been demonstrated that the function of the IB is affected by physical activity and that heavy physical exercise, like marathon running or ultra-marathon running, results in massive endotoxemia [10,11,12]. It has also been shown that even an acute or short-term intake of specific foods or food compounds, like saturated fats or fructose and sucrose [13,14,15], may impair IB function and increase bacterial endotoxin levels in humans, thereby adding to the development of so-called post-prandial endotoxemia [16]. However, whether sugars consumed in their natural matrix state, like fruits or fruit juices, affect IB function and the translocation of bacterial wall compounds in the same way, has not yet been clarified in humans. Indeed, in contrast to most soft drinks, fruit juices contain a variety of secondary plant metabolites that are absorbed in the intestine and are also metabolized by intestinal cells. For instance, the results of in vitro and ex vivo studies have suggested that polyphenols, like those found in apple juice, are absorbed and metabolized [17,18], and may thereby affect inflammatory processes and even DNA damage [19,20]. Information about the combinatory effects of sugar consumption and exercise on the IB is limited.

Thus, it was the purpose of this study to analyze the combinatory effects of sugars consumed in their natural matrix (diluted cloudy apple juice with a high content of polyphenols (test drink)) or in water (placebo), in comparison to water, and in combination with exercise, on the markers of the IB. In the first study, this was analyzed after moderate exercise in a placebo-controlled, blinded, crossover design (study A). In addition, the effects of the same drinks were tested in a second study after high-volume endurance exercise (study B). Here, the finishers of an ultra-marathon were divided into groups and consumed the test drink, the placebo, or water directly after finishing.

## 2. Methods

### 2.1. Participants

#### 2.1.1. Study A

All the participants in this study were recruited from local running groups or from courses at the German Sport University Cologne. This study was carried out between December 2021 and October 2022. The sample size was calculated with G*Power and a total of 24 participants were considered sufficient for enrollment (Gpower, Version 3.1.9.2, Düsseldorf, Germany), based on the effect sizes of previous studies [2].

A total of 24 men and women were enrolled in this study, of whom a total of 17 participants completed this study. Participants dropped out of this study due to family reasons (*n* = 2), illness (*n* = 2), or unspecified reasons (*n* = 3). None of the participants smoked, suffered from chronic or acute diseases, or was overweight (BMI between 18.5 kg/m^2^ and 24.9 kg/m^2^). All the participants followed an omnivorous diet. None of the participants reported drinking more than a moderate amount of alcohol (no more than one glass of beer or wine, so that a quantity of >10 g/day for women and >20 g/day for men was not exceeded). They were healthy, generally physical active, but not specifically endurance-trained. In addition, the subjects had to be between 18 and 35 years old, have no acute injuries or chronic illnesses, take any medications, and have a running time for 10,000 m of <55 min for women and <50 min for men. High-performance athletes with low body fat and individuals trained specifically for national and international competitions were also excluded. Before the subjects were included in this study, they were informed about the study design and objectives and had to give their written consent to participate. All personal data collected were anonymized and in compliance with data protection regulations in Germany. An overview of the subjects’ characteristics is given in Table 1.

#### 2.1.2. Study B

Ultra-marathon (TorTour de Ruhr^®^) participation required a medical sports examination conducted less than 6 months before the race and that confirmed the physical resilience of the athlete. The inclusion criteria for study participation were as follows: male/female; endurance athlete and 160.9 km or 230 km participant in the TorTour de Ruhr^®^ 2022; no previous injuries; and the ability to understand this study’s procedure and to give informed consent. The exclusion criteria included the following: participants in the 100 km race (owing to organizational restrictions); nicotine consumption; diseases of the intestine; blood clotting disorders or intake of blood-thinning medications; acute or chronic vascular (blood flow) disorders; cardiovascular, metabolic, or autoimmune diseases; and non-consenting subjects. An overview of the subjects characteristics is given in Table 2.

All the subjects received information about this study’s content and the use of the data and provided their written consent.

Both study parts were approved in their entire study designs by the local ethics committee of the German Sports University Cologne in October 2021 (reference number: 171/2021) and registered with the DKRS under trial No. DRKS00027860.

### 2.2. Study Design

#### 2.2.1. Study A

This study was designed as a randomized, double-blind trial with a crossover design. To ensure accuracy, there was a washout period of at least 2 weeks between intervention phases to account for possible training adaptations and the effects of the study beverages. The intervention consisted of a one-hour endurance run at an intensity of 80% of the individual anaerobic threshold (IAT). To determine the IAT, a field incremental exercise test was conducted. The test started at a velocity of 2.4 m/s, increasing by 0.4 m/s every 5 min. Each increment was followed by a 30 s rest period that allowed for capillary blood sampling, enabling the analysis of the blood lactate concentration [21]. The test continued until volitional exhaustion was reached, determined when at least two of the following three criteria were met: (1) reaching the age-predicted maximum heart rate, (2) a blood lactate concentration of 8 mmol/L or higher, and (3) a rating of 18 or higher on the Borg Scale [22]. The analysis of the IAT included a mathematical calculation of the lactate concentration with the Dmod method [23]. This method determines the maximum distance between a third-degree polynomial curve—representing the lactate data from the incremental test for each participant—and the line connecting the first point of lactate rise (increase of ≥0.4 mmol/L between two increments) to the final measured lactate concentration.

The endurance run was carried out three times, with the test drink, placebo, and water supplementation. The study beverages were ingested immediately after the run. Blood samples were taken at defined time points (T0 (pre-exercise (pre-ex)), T2 (post-exercise (post-ex)), T3 (+80 min), T4 (+120 min), and T5 (+180 min)) throughout the study day and a further blood sample was taken on the following day. Figure 1 displays the entire study design.

On the evening before each of the three study days, the 17 healthy subjects consumed a standard dinner. The meal consisted of pasta in a tomato cream sauce with zucchini and mozzarella with the macronutrients, i.e., the energy derived from carbohydrates at 56 E%, fat at 30 E%, and protein at 14 E%, based on the recommendations of the German, Austrian, and Swiss nutrition societies (DACH) (see Table 3 for meal composition). Protein levels were adjusted for the higher needs of athletes [24]. The energy content of the meal was based on the ‘normal’ energy intake of each participant, taking gender, physical activity, and age into consideration. To assess the ‘normal’ energy intake of the participants, two independent 24 h recalls (for one weekday and one weekend day) were performed by a trained nutritionist. The nutritional intake and standardization of the meal were calculated via the computer software EBISpro (Version 2011, Willstätt, Germany).

The results of the 24 h recalls and the values of the energy and macronutrient composition of the standard meal are shown in Appendix A.

On each study day, all study participants arrived at the German Sport University Cologne in a fasting state in the morning, having fasted for at least 12 h since their last meal. Upon their arrival, a first blood sample was taken (T0). Subsequently, all the subjects received a standardized breakfast, which consisted of an amount of oatmeal adjusted to their nutritional needs for their body weight determined by 2 independent 24 h recalls, 5 g of honey or agave syrup, a banana, and hot water (ad libitum). After a defined resting period of one hour, all study participants completed a one-hour endurance run at an intensity of 80% IAT. During the defined one-hour endurance run, their heart rates (HR) were recorded using heart rate monitors. Immediately after the completion of the run, blood lactate levels were measured for the blood obtained from the capillary blood of the ear for load control, and a second blood sample was collected (T2, post-ex). Subsequently, the drinks (500 mL water, placebo, or test drink) were consumed by the participants within 5 min. During the post-exercise period, additional blood samples were collected after 80 min (T3, +80 min), 120 min (T4, +120 min), 180 min (T5, +180 min), and after 21 h (T6, +24 h), following the completion of beverage consumption. The subjects were not allowed to engage in additional physical activity during this time interval, and they were not permitted to consume any meals until after the T5 (+180 min) blood sample was taken. After the completion of the study day (+24 h blood sampling on the following day), there was a washout period of at least 14 days before the study procedure was repeated with the remaining study drinks on a sample-specific basis.

#### 2.2.2. Study B

The study participants competing in the TorTour de Ruhr^®^ ultra-marathon (230 km and 160.9 km distances) (*n* = 30) were randomly divided into three groups (*n* = 10), and ingested the study beverages immediately after the run. Blood samples were taken at defined time points (T0 (pre-exercise (pre-ex)), T2 (post-exercise (post-ex)), T3 (+120 min), and T4 (+180 min)). Blood samples were stored at 4 °C until transportation to the analysis facility. All samples were transported within a time frame of a maximum of four hours, and were either measured immediately in the laboratory or stored at −80 °C for further analysis. Figure 2 displays the entire study design.

### 2.3. Study Beverages

The study beverages were either 500 mL water (as the negative control), a sugary iso-sweet and iso-caloric control drink (placebo), or diluted cloudy apple juice (test drink). Volvic water (Danone Deutschland GmbH, Frankfurt am Main, Germany) was used as the negative control because it is particularly low in nutrients and minerals. The sugary comparative drink (placebo) contained sugars (fructose, glucose, and sucrose) in the same proportion as the diluted cloudy apple juice, as well as the aromas and acids typical of apple juice, and was isotonic. Compared to the diluted cloudy apple juice, the placebo contained no polyphenols or secondary plant substances. The diluted cloudy apple juice had a fruit juice content of 60% naturally cloudy apple juice with the highest possible polyphenol content (see Table 4).

#### 2.3.1. Density and Brix

The density of the juice was measured using a bending oscillator (Anton Paar, Seelze, Germany). The corresponding Brix value was calculated using the density table of the IFU (International Fruit and Vegetable Juice Association) No. 01a (Rev. 2005), and should correspond to the corrected Brix value (Bx. Corr.) [25].

#### 2.3.2. Total Acidity

The total acidity was determined titrimetrically, based on method no. 03 (Rev. 2017) of the IFU (International Fruit and Vegetable Juice Association) [26].

#### 2.3.3. Determination of the Mineral Content Using ICP-OES (Optical Emission Spectrometry with Inductively Coupled Plasma)

The mineral content of the juice was determined after microwave digestion using ICP-OES. The minerals were determined analogously to the ASU method L.00.00-19/1 [27]. Prior to this, the juice samples were prepared using microwave digestion, which was carried out analogously to ASU method L.00.00-144 [27].

#### 2.3.4. Enzymatic Determination of the Sorbitol Content

The sorbitol content of the juice was automatically determined enzymatically and kinetically using a test kit from Boehringer Mannheim/r-biopharm (Darmstadt, Germany).

#### 2.3.5. The Sugars Present in the Juice Were Characterized and Quantified Using HPLC-RID (High-Performance Liquid Chromatography with Refractive Index Detector)

The sugar types—glucose, fructose, and sucrose—were also quantified using HPLC-RID based on method no. 67 (Rev. 2005) of the IFU (International Fruit and Vegetable Juice Association) [28].

#### 2.3.6. Total Phenol Content, Determined with Folin–Ciocalteu

The Folin–Ciocalteu reagent was used to determine the total phenol content of the juice, according to the method of Prior et al. (2005) [29].

### 2.4. Blood Analysis

Bacterial endotoxin levels were assessed by using a limulus amebocyte lysate assay (Charles River, Ecully, France), as detailed previously in [30].

The protein concentrations of the intestinal fatty acid-binding protein (i-FABP) and soluble cluster of differentiation 14 (CD14) in the serum of the participants were determined by using commercially available ELISA kits (Bio-Techne Corp., Minneapolis, MN, USA) according to the manufacturer’s instructions. The serum concentrations of interleukin 6 (IL-6) protein were assessed via human IL-6 ELISA Kit (R&D Systems, Minneapolis, MN, USA).

The skeletal muscle-specific creatine kinase (CK) concentration in the serum was determined as a marker for skeletal muscle damage using a COBAS h 232 point-of-care system (Roche Diagnostic Systems, Rotkreuz, Switzerland).

### 2.5. Statistical Analysis

The data were analyzed using SPSS (IBM SPSS Statistics 26.0, Ehningen, Germany) and R version 4.3.0 [31]. Linear mixed-effects regression models (LME) with a random intercept, and fixed effects for the group (water vs. placebo vs. juice), time (before vs. post-exercise vs. +180 min after exercise), and their interaction (group × time), were used to analyze the effects of the intervention over time on the IL-6 protein in the serum. As the assumption of a linear regression was not met for the IL-6 protein, a robust linear mixed regression was conducted for the IL-6 protein levels (R-package: robust mm) [32]. To review the robustness of the results, we repeated the analyses, adjusting for possible covariates of sex, age, and BMI (see Appendix A). The effect size partial η^2^ is reported for the group effects, time effects, and group × time interaction. The measurement parameters that were collected in relation to the one-hour running load were tested for normal distribution with the Kolmogorov–Smirnov test. The raw data for CK were transformed into their natural logarithms (ln) before data analysis. Subsequently, the CK blood concentration was analyzed for time (pre-exercise vs. post-exercise), group (water vs. placebo vs. apple juice spritzer), and time × group interaction, with a repeated measures ANOVA. The running-related parameters of HR and lactate were also analyzed with one-way ANOVA for the three different study beverage conditions. The reported *p*-values of the LME were two-tailed, the *p*-values of the ANOVA were one-tailed, and both had a significance level of α < 0.05.

The data for endotoxin concentrations, i-FABP, and CD14 are presented as the mean ± SEM. The test for normality was analyzed using the Shapiro–Wilk normality test. Grubb’s test was performed before the statistical analysis to identify outliers. To assess the effects between different time points, a paired *t*-test was used. An ordinary one-way ANOVA with Dunnett’s and Turkey’s multiple comparison test was used to determine statistically significant differences between the interventions. All the data were analyzed with GraphPad Prism software version 10 (GraphPad Prism Software Inc., Solana Beach, CA, USA). Significance was considered for a *p*-value <0.05.

## 3. Results

### 3.1. Study A

#### 3.1.1. Effects of One-Hour Running Load on CK, Lactate, and HR

For the analysis of CK blood concentration, 16 subjects were included because it was not possible to take one blood sample at T6 (+21 h) due to the time constraints of one subject. A repeated measures ANOVA determined that the mean CK blood concentrations were statistically significantly different before and after the one-hour running load (F(1, 45) = 50.36, *p* < 0.001, ηp^2^ = 0.528) (Figure 3A,B). Among the study beverage conditions, the water, placebo, and test drink CK values were not statistically significantly different (F(2, 60) = 0.289, *p* = 0.750) (Figure 3C). Furthermore, no significant time x group interaction could be identified (F(2, 45) = 0.41; *p* = 0.667).

The mean lactate blood concentration (Figure 3B) showed no statistically significant difference between the three beverage conditions (F(2, 48) = 0.566, *p* = 0.57). Similarly, there were also no statistically significant differences for the HR values (Figure 3A (F(2, 48) = 0.114, *p* = 0.892).

The results shown in Figure 3 demonstrate that the running load was comparable for all three interventions (water, placebo, test drink) among the individuals (Figure 3A,B), and the running intensity was sufficient to result in moderate skeletal muscle damage (Figure 3C).

#### 3.1.2. Effects of One-Hour Moderate Running Load on Endotoxin, i-FABP, and CD14

To analyze the impact of the running load on the IB function, the serum concentrations of endotoxin, i-FABP, and CD14 protein were measured for the blood samples collected at T0 and T2. The results are shown in Figure 4.

As shown in Figure 4, the one-hour running activity at moderate intensity resulted in a statistically significant increase in LPS, i-FABP, CD14, and IL-6 in the serum of our volunteers.

#### 3.1.3. Effects of Study Beverages on Time-Dependent Change in Serum Concentrations of Bacterial Endotoxin as Well as in i-FABPIL-6 and CD14 Protein in Serum after Run

To analyze how the intake of the respective beverages after running affects the time-dependent concentration of the target biomarkers for intestinal barrier function and inflammation, the volunteers consumed the test beverages directly after their runs. Blood samples were collected before the consumption of the respective drinks, after 120 min, and after 180 min. The results are shown in Figure 5.

In Figure 5, it is visible that after exercise the serum levels of endotoxin (Figure 5A), i-FABP (Figure 5B), and IL-6 protein (Figure 5D) decreased in all groups at a comparable rate. The increase in the respective serum concentrations induced by the running activity returned to pre-exercise levels after 180 min. No statistically significant impact of the different drinks could be observed. For CD14, significant differences could be observed between the placebo group and the test drink group, with the latter being significantly higher than after intake of the placebo.

### 3.2. Study B

Of the 30 participants, only 10 finished the race and ingested the study beverages immediately after the run (water, *n* = 3; placebo, *n* = 3; test drink, *n* = 4). Blood samples were taken before the run, immediately before consumption of the study beverages, and 120 min and 180 min post-exercise.

#### 3.2.1. Effects of the 160 and 240 km Runs on Serum Concentrations of Bacterial Endotoxin and IL-6 Protein in Serum

To analyze the impact of the ultra-marathon on the IB and the inflammatory status of the participants, blood samples were collected from the finishers and serum concentrations of bacterial endotoxin and IL-6 protein were determined. The results are presented in Figure 6.

As shown in Figure 6, the ultra-marathon results in a strong significant increase in serum bacterial endotoxin (Figure 6A) and serum IL-6 protein concentrations (Figure 6B). Serum concentrations of IL-6 exceeded the serum concentrations detected after moderate running activity (Figure 4) more than 20-fold.

#### 3.2.2. Effects of the Study Beverages on Time-Dependent Serum Concentrations of Serum Endotoxin after the Run

To evaluate the impact of the three respective beverages administered after the ultra-marathon on the time-dependent serum concentration of bacterial endotoxin, the volunteers consumed the beverages directly after finishing the run. Blood samples were collected before consuming the respective drinks, and after 120 min and 180 min. The results are shown in Figure 7.

After the ultra-marathon, serum endotoxin levels decreased very quickly. This could be observed in the volunteers consuming water, but also in those consuming the test drink. However, in the volunteers consuming the placebo, the AUC over the 180 min testing period was significantly higher compared to the water group.

## 4. Discussion

In our studies, we investigated the impact of sugars and the matrix ingredients of cloudy apple juice (FJM) on the IB after moderate and intensive physical activity.

Study A was conducted using a crossover design. As shown in Figure 3, the training load for our volunteers was comparable in all three interventions. Moreover, the intensity of the running activities resulted in moderate skeletal muscle damage, indicated by the increased CK serum concentrations. Therefore, well-defined baseline conditions were established to compare the impact of our three different drinks on the IB after exercise in the crossover design. In Figure 4, it is evident that a moderate training load of 80% of IAT for a duration of 60 min resulted in a significant increase in serum endotoxin, i-FABP, and IL-6 protein. For the bacterial endotoxin, this is in agreement with the published data [33]. For i-FABP, an increase in serum protein has been shown before after 90 min of running activity [11]. Our data demonstrate that moderate exercise has an impact on the IB and results in an induction of inflammation, as indicated by the increase in IL-6 protein levels in the serum. In all three groups, the levels of serum endotoxin and IL-6 protein quickly returned to the baseline levels within a period of 180 min after exercise (Figure 5). To our knowledge, this fast decrease has not been described before. Immediately after completing the running exercise in study A, the volunteers received their respective drinks. In previous investigations it has been demonstrated that the uptake of sugars, especially fructose, has an impact on the IB and results in leakiness [14,15]. Therefore, it was one of our hypotheses that the uptake of sugar-containing drinks after exercise may harm the IB in an additive manner, and may also impact the time-dependent regeneration of the IB after setting the stressor. As shown in Figure 5, neither the placebo drink nor the test drink seems to have affected the time-dependent decrease in our marker’s bacterial endotoxin, i-FABP, or IL-6. For the i-FABP and IL-6 protein, no statistically significant difference could be observed compared to water. For the endotoxins, a statistically significant difference could be observed between the placebo and water after 180 min. However, whether this effect is of physiological relevance is questionable. No differences could be observed between water and the test drink, demonstrating that the test drink acts, independent of its sugar content, comparably to water (Figure 5D).

More complex are the effects observed for CD14. Soluble CD14 detected in serum is known to be a nonspecific marker of monocyte activation, independent of bacterial endotoxin [34]. The observed increase in the CD14 protein induced by exercise is in line with reports from the literature [35]. In Figure 5C, it is clearly visible that the concentration of soluble CD14 in the serum differed between the respective groups 180 min after exercise. A time-dependent decrease in the CD14 protein could be observed in the test drink group, whereas after drinking the placebo, a statistically significant increase in the CD14 protein was detectable. We speculate that the pattern for CD14 is different than that for IL-6, endotoxin, and i-FABP because CD14 seems to be more affected by exercise than by endotoxin. Whether the increase in the CD14 protein in the test drink group indicates an adverse or beneficial effect is unclear. However, in the water group, the CD14 protein levels also did not decrease after 180 min. The observed increase in the CD14 protein serum concentrations 180 min after exercise could be interpreted as an indirect marker for leukocytosis [35,36]. Exercise has been shown to increase circulating neutrophil and monocyte counts and reduce circulating lymphocyte counts during recovery after physical activity. This redeployment of effector lymphocytes is an integral part of the physiological stress response to exercise. Some evidence suggests that reduced immune cell function may coincide with rates of illness after exercise. Therefore, a longer detectable leukocytosis could be interpreted as an increase in immune cell function to protect against infections after exercise.

In study B, we tried to analyze the impact of our test drinks in a situation where the IB was already strongly stressed. After ultra-marathon runs, it has been demonstrated that the IB function is massively disturbed [37]. Therefore, we recruited 30 participants in an ultra-marathon run over 160–230 km, but because only 10 of these participants finished the ultra-marathon, this investigation must be characterized as a pilot study.

As shown in Figure 6, the finishers’ serum endotoxin concentrations and serum IL-6 levels increased dramatically during the race. This agrees with the published data [10] and indicates a massively leaky IB and enormous systemic inflammation after finishing the race. The serum levels of LPS exceeded the levels detected after moderate exercise for one hour at 80% IAT by 3-fold. The serum levels of IL-6 protein after finishing the ultra-endurance race were 20-fold higher than after the one-hour run at 80% IAT. Remarkably, our volunteers who drank water immediately after finishing showed a fast decrease in endotoxins in the first 120 min after arrival (Figure 7). It is remarkable that endotoxin serum levels returned to normal concentrations so fast after an ultra-marathon. This observation is in agreement with the data from Jeukendrup et al., who collected blood samples from 29 athletes before, immediately after, and 1, 2, and 16 h after a long-distance triathlon to measure endotoxin, tumor necrosis factor-alpha, and IL-6 levels [38].

Even more remarkable is the impact of our beverages. In Figure 7, it is clearly visible that the consumption of the placebo, containing free sugars, resulted in significantly higher bacterial endotoxin concentrations in these participants after 120 min, compared to the water group. Also, the AUC over the 180 min for this group is significantly higher than for the water group. This might be an indication that the uptake of free sugar-containing drinks in a situation with an already enormously stressed IB may negatively impact its regeneration. Such effects, to our knowledge, have not been described in previous studies, but agree with findings that have demonstrated that free sugars have a negative impact on the IB [14,15] under resting conditions. In study A (Figure 5), a comparable effect could not be observed, indicating that this impact may be more pronounced after high volume performance intensity.

A surprising result is our observation that among our volunteers who consumed our test drink, which contained the same amount of sugar and had a composition like that of the placebo but was embedded in the FJM, the measured serum LPS levels are not different to those for water after 120 min. This is also the case for the AUC over the 180 min period. This observation indicates that the fruit juice matrix may have counteracted some of the negative impact of the sugars on the IB in the period after running.

### Limitations and Outlook

This study introduces novel insights into the effect of physical activity on intestinal barrier function; however, when interpreting the presented data, some limitations need to be considered.

In study A, the volunteers’ dietary intake was only standardized 12 h before we started the intervention (standardized dinner), and on the day of the intervention, until the 3 h post-running sampling point. A longer period of nutrition standardization would be preferable, and may result in the clearer effects of our drinks on IB functionality, as has been shown before by [14,15]. However, in this study this could not be realized for organizational reasons. Also, in study A, the majority of the participants were male.

In study B, nutrition could not be standardized at all because of the racing conditions and ethical reasons surrounding the ultra-marathon run. Also, after finishing, a standardization of nutrition would have been preferable and may have led to clearer results. However, this was also not possible for ethical reasons regarding the individuals who wanted to finish an ultra-marathon competition.

Also, another limitation was the high drop-off rate of our ultra-marathon runners. Climate conditions during the race worsened and were a major reason for these unexpectedly high drop-off rates. The limited number of finishers resulted in a small sample size for the following investigation, which limits the scientific quality of our data. However, the results are interesting and we believe that similar effects could be also observed in marathon finishers. Therefore, we plan to extend this study in a similar setting during a regular marathon with a much higher number of participants under better-standardized conditions.

## 5. Conclusions

In our study, we show that even moderate endurance exercise results in a slight but significant increase in serum endotoxin levels, indicating a leakiness of the IB. The IB regenerates fast after moderate exercise. The consumed drinks did not affect the kinetics of IB regeneration, but had an impact on the CD14 serum concentrations, indicating that the FJM of apple juice may have beneficial effects on the immune system after exercise. An ultra-marathon results in a very strong increase in serum endotoxins, indicating IB dysfunction. Here also, IB function regenerated very fast after the consumption of water, whereas the intake of plain sugar (placebo) seems to have had a negative impact on IB regeneration, which was diminished by the presence of the FJM in the test drink. Because the uptake of sugars is necessary for recovery after intensive exercise, the results of this study provide evidence that the negative impact of plain sugars on the IB may be compensated for by the ingrediencies of the FJM. Diluted apple juice is a well-known drink for rehydration after physical exercise. Beside these already well-known beneficial effects, it may also have positive effects on the IB and the immune system after exercise.

## Figures and Tables

**Figure 1 nutrients-16-01353-f001:**
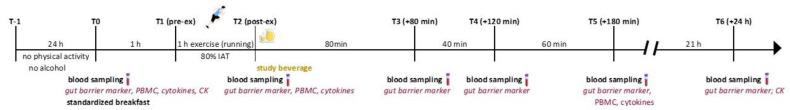
Study design of study A. CK = creatine kinase; IAT = individual anaerobic threshold; PBMC = peripheral blood mononuclear cells; pre-ex = pre-exercise; post-ex = post-exercise.

**Figure 2 nutrients-16-01353-f002:**
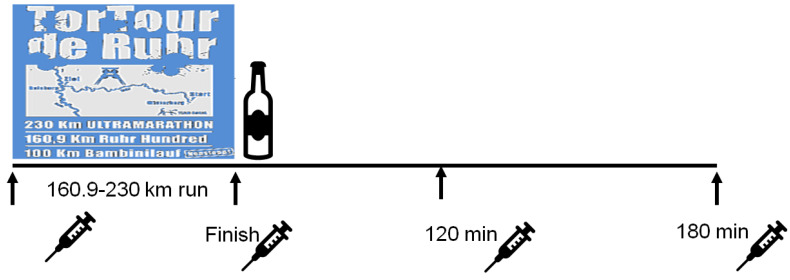
Study design of study B.

**Figure 3 nutrients-16-01353-f003:**
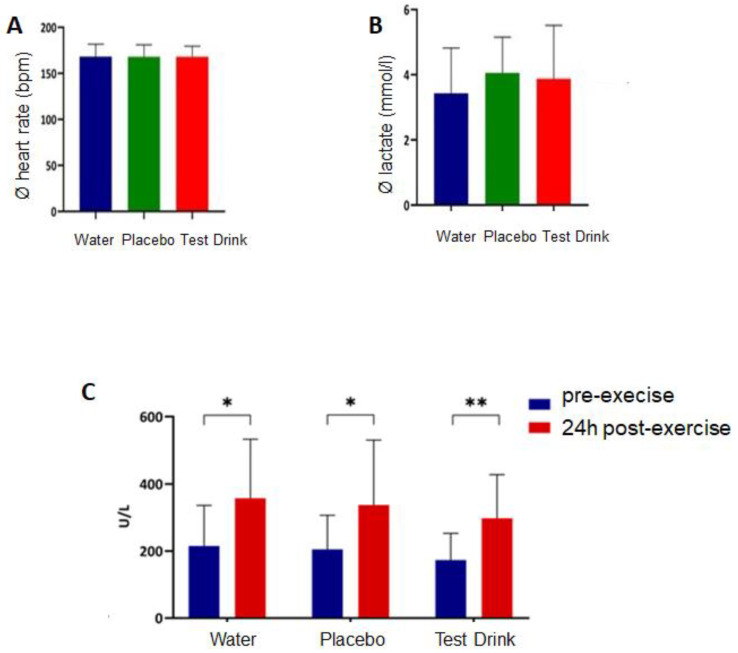
Effect of the one-hour running load on heart rate (HR), lactate, and creatine kinase (CK). Shown are the mean values of the three runs. (**A**) Average heart rate, (**B**) average lactate serum concentrations at the end of the run, (**C**) change in CK concentrations (delta) for all three runs. Mean comparisons are based on unpaired i-tests and ANOVA with repeated means (water, *n* = 15; placebo, *n* = 13; test drink, *n* = 14; *, *p* < 0.05; **, *p* < 0.01).

**Figure 4 nutrients-16-01353-f004:**
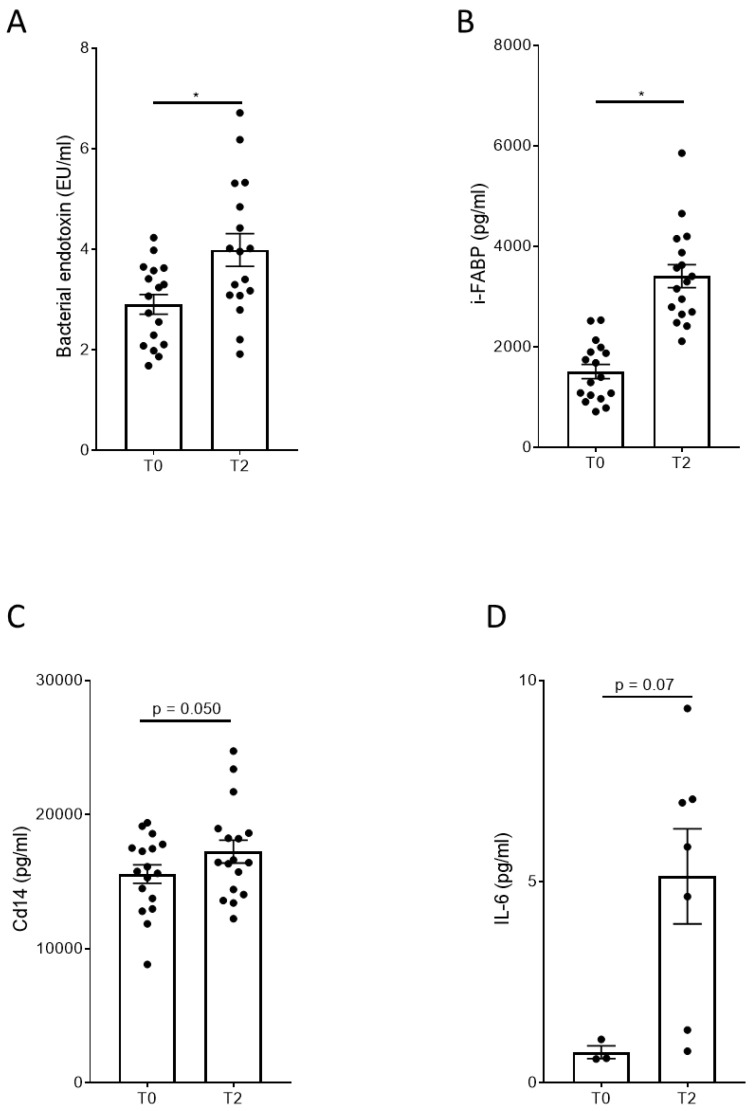
Effect of the physical activity on the markers of intestinal permeability and inflammation in the serum of participants. (**A**) Bacterial endotoxin, (**B**) i-FABP, (**C**) CD14 protein, and (**D**) IL-6 protein levels before (T0) and after (T2) the physical activity in the serum of participants. Values are means ± SEM; *n* = 17 except for (**D**), *n* = 3–7; *, *p* < 0.05. i-FABP = intestinal fatty acid-binding protein; CD14 = cluster of differentiation 14; IL-6 = interleukin 6. Results show that a moderate running intensity significantly increases serum markers for the functionality of the intestinal barrier.

**Figure 5 nutrients-16-01353-f005:**
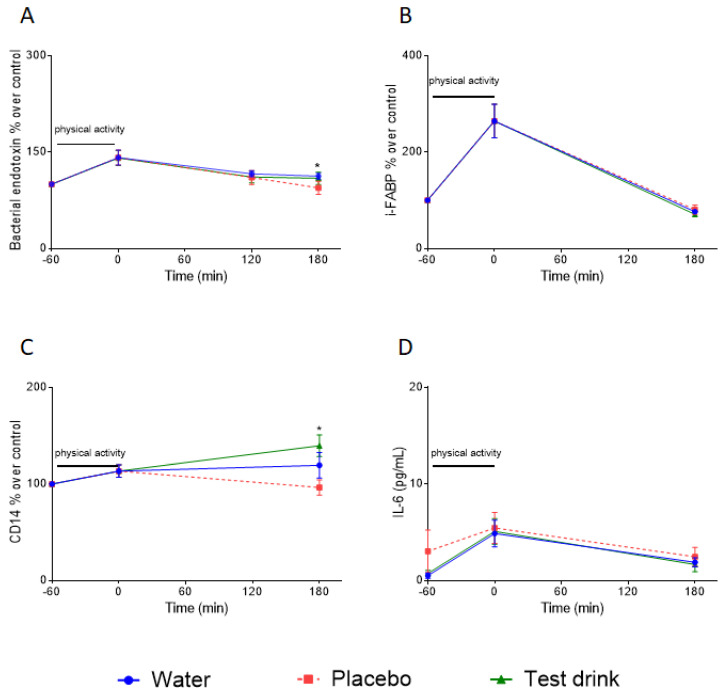
Effect of different beverages after physical activity on markers of intestinal permeability and inflammation in serum of participants. (**A**) Bacterial endotoxin concentration, (**B**) i-FABP, (**C**) CD14, and (**D**) IL-6 protein concentration after 120 and 180 min in serum of participants following 1 h of physical activity (−60–0 min) at 80% IAT. Values are means ± SEM; *n* = 17; for (**A**), * indicates *p* < 0.05 for water vs. placebo after 180 min; and for (**C**), * indicates *p* < 0.05 for placebo vs. test drink. i-FABP = intestinal fatty acid-binding protein; CD14 = cluster of differentiation 14; IL-6 = interleukin 6.

**Figure 6 nutrients-16-01353-f006:**
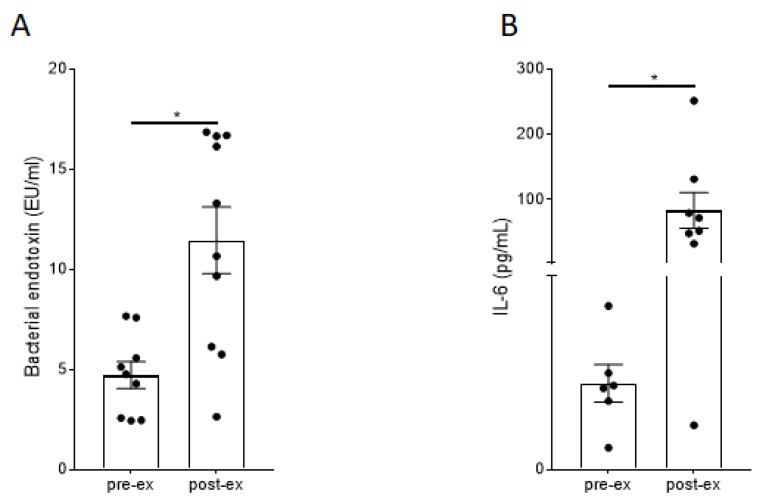
Effect of the ultra-marathon on bacterial endotoxin levels and IL-6 protein in the serum of the participants. (**A**) Bacterial endotoxin and (**B**) IL-6 levels in the serum of the athletes. Values are the means ± SEM. For (**A**,**B**): *n* = 6–10; *, *p* < 0.05.

**Figure 7 nutrients-16-01353-f007:**
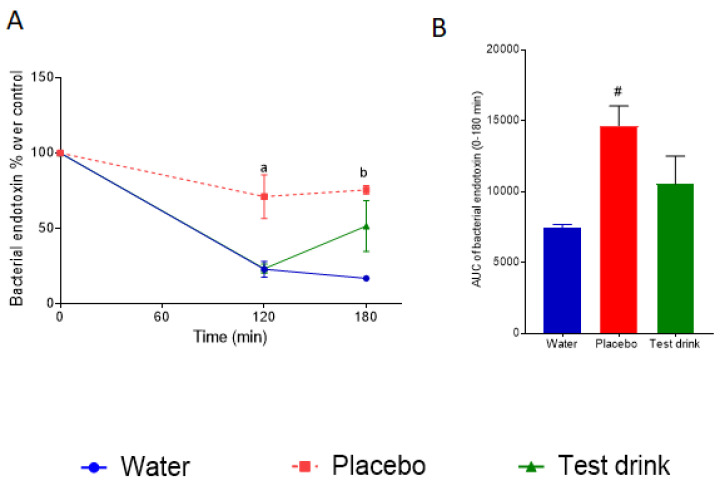
Effect of different beverages after finishing an ultra-marathon on serum concentration of bacterial endotoxin. (**A**) Bacterial endotoxin levels and (**B**) AUC (0–180 min) of bacterial endotoxin levels of participants. Values are means ± SEM. For (**A**,**B**): *n* = 3–4; ^a^ *p* < 0.05; placebo vs. water, water vs. test drink; ^b^
*p* < 0.05; placebo vs. water; ^#^
*p* < 0.05 placebo vs. water. IL-6 = interleukin 6; AUC = area under the curve.

**Table 1 nutrients-16-01353-t001:** Overview of subjects characteristics. Shown are mean results ± standard deviation (SD).

Sex	Male (14), Female (3)
Age (years)	26 ± 4
Height (cm)	178.94 ± 7.28
Body mass (kg)	70.94 ± 8.17
BMI (kg/m^2^)	22.12 ± 1.80

**Table 2 nutrients-16-01353-t002:** Anthropometric data, finish time, and nutrition details for this study’s participants. All data are presented as mean ± standard deviation.

Anthropometry							
Age (years)		Height (cm)		Body Mass (kg)		Body Mass Index (kg/m^2^)	
160.9 km	230 km	160.9 km	230 km	160.9 km	230 km	160.9 km	230 km
50.20 (7.98)	50.14 (9.74)	174.80 (4.62)	177.43 (8.74)	71.14 (4.18)	71.89 (11.30)	23.30 (1.29)	22.79 (2.93)
Finish time (hours)							
160.9 km				22.56 (3.31)			
230 km				32.28 (2.95)			

**Table 3 nutrients-16-01353-t003:** Composition of the standardized meal served to the participants on the evening before the intervention day.

Meal Components	Energy (kcal/100 g)	Carbohydrates (g/100 g)	Fat (g/100 g)	Protein (g/100 g)	Fiber (g/100 g)
Egg-based pasta	352.3	68.3	2.8	12.3	5.0
Zucchini	19.1	2.0	0.4	1.6	1.1
Onion	28.0	4.9	0.3	1.3	1.8
Olive oil	881.7	0.2	99.6	-	-
Tomato	17.4	2.6	0.2	0.9	0.9
Cream (10% fat content)	116.6	3.3	10.0	3.1	-
Mozzarella	254.8	-	19.8	19.0	-
Salt	-	-	-	-	-

Values were determined using the software EBISpro.

**Table 4 nutrients-16-01353-t004:** Ingredients in diluted cloudy apple juice (test drink) and matched sugary control drink (placebo) per liter.

Parameter	Diluted Cloudy Apple Juice (60% Juice Content)	Placebo
Brix corr.	6.72	6.72
Glucose [g/L]	13.4	13.4
Fructose [g/L]	35.0	35.0
Sucrose [g/L]	9.1	9.1
Total sugar [g/L]	57.5	57.5
Sorbitol [g/L]	2.20	2.20
L-malic acid [g/L]	3.36	3.36
Potassium [g/L]	0.59	0.59
Apple flavor [ml/L]	---	4.70
Total polyphenol content [mg/L]	554.4	---

## Data Availability

The raw data can be viewed upon request.

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
