# Peer review of "Effect of Sugar- and Polyphenol-Rich, Diluted Cloudy Apple Juice on the Intestinal Barrier after Moderate Endurance Exercise and in Ultra-Marathon Runners"

_nutrients, 2024, doi:10.3390/nu16091353_

Round 1
Reviewer 1 Report
Comments and Suggestions for Authors
1. Affiliations: department instead of Dept.
2. Indicate corresponding author and add e-mail address for corresponding author
3. Abstract: write LPS and IAT in full, add info for number of subjects, gender, age range
4. Introduction: instead of increase regeneration, replace with aid in recovery (after) and provide energy (during); check if the references 1 and 2 cover this, there are several extra spaces between words
5. Methods: provide all details for calculation of sample size, moderate amount of alcohol describe with glasses/portions instead of with grams of etOH
6. Table 2: needs complete correction, regarding title, content, number, parameters
7. Provide all details for D-mod method
8. L 126-31: needs correction; 24h recall does not serve to define portion size, but to determine usual intake, so change in description is needed here, also, provide numbers for referent values, and actual intake of energy and macronutrients; the meal was universal or individually tailored? Also, this is just one meal, and not entire daily intake, so indicated % kcal do not apply; this section requires extensive improvement, and add info for food composition data used; all this apply for L 135-137; what is ‘’2 independent 24-hour’’, and what are the actual numbers for intake?
9. Error! Reference source not 159 found; should be corrected
10. There are several mentioning of ethics approval; condense and elaborate in single paragraph
Author Response
We thank the reviewer for its positve and helpful comments . Here is our point by point response:
Affiliations: department instead of Dept.
Done
Indicate corresponding author and add e-mail address for corresponding author
Done
Abstract: write LPS and IAT in full, add info for number of subjects, gender, age range
Done
Introduction: instead of increase regeneration, replace with aid in recovery (after) and provide energy (during); check if the references 1 and 2 cover this, there are several extra spaces between words
We have checked the references , we replaced the wording and we corrected the extra spaces
Methods: provide all details for calculation of sample size, moderate amount of alcohol describe with glasses/portions instead of with grams of etOH
Done
Table 2: needs complete correction, regarding title, content, number, parameters
Done
Provide all details for D-mod method
Done
L 126-31: needs correction; 24h recall does not serve to define portion size, but to determine usual intake, so change in description is needed here, also, provide numbers for referent values, and actual intake of energy and macronutrients; the meal was universal or individually tailored? Also, this is just one meal, and not entire daily intake, so indicated % kcal do not apply; this section requires extensive improvement, and add info for food composition data used; all this apply for L 135-137; what is ‘’2 independent 24-hour’’, and what are the actual numbers for intake?
We corrected the mentioned paragraph of the paper according to the suggestions and remarks of the reviewer. Also we provide a new table and we ad supplemental material
Error! Reference source not 159 found; should be corrected
There are several mentioning of ethics approval; condense and elaborate in single paragraph
Reviewer 2 Report
Comments and Suggestions for Authors
The authors investigated the effect of apple juice on the intestinal barrier after exercise. I think it is interesting and important topic. For the selection and exclusion of participants, the paper has done a careful description, other experimental methods are also described in detail, which will be conducive to do repeat study. The structure of the paper is clear, but there are the following problems:
1. It is better to do correlation analysis between total polyphenol dosages and levels of inflammatory factors and endotoxin?
2. For experiment design, adding new apple polyphenol group in the study would better confirm whether polyphenols or other active ingredients are involved in the anti-inflammatory and endotoxin-lowering effects.
3. In Figure 5C, why the IL-6 level (pre) of placebo juice group is higher than other two groups?
Comments on the Quality of English LanguageThere are some typography and small language errors
Author Response
We thank the reviewer for its positive remarks to our manuscript and for the questions on comments. Here comes our point by point reply.
- It is better to do correlation analysis between total polyphenol dosages and levels of inflammatory factors and endotoxin?
In our study we have used a polyphenol rich apple Juice. We know the exact polyphenol content but not the exact composition. Moreover, in this first die we did not focus specifically on the polyphenpols. We focused on the general fruit juice matrix. Therefore, we think that a correlation analysis of the total polyphenol dosage to the measured parameters will not result in other results like we showed, because the polyphenole content of the placebo was zero. We agree completely that it will be interesting in future Studies to do does responses between the polyphenol doses and the detected effects. Especially this will be interesting with respect to the CD14 data. We thank the reviewer for this suggestion.
- For experiment design, adding new apple polyphenol group in the study would better confirm whether polyphenols or other active ingredients are involved in the anti-inflammatory and endotoxin-lowering effects.
We agree completely for future studies. See also our answer to your comet 1. Like mentioned in the limitations of our paper we are aware about this and this will be an aim of a future study with is already in preparation.
- In Figure 5C, why the IL-6 level (pre) of placebo juice group is higher than other two groups?
Yes, you are right. The reason is that one of the participants has a strongly elevated level of IL6 at this time point and treatment group. The reason is unknown. This is also the reason for the high error bar of IL6. However, the alternative would have been to eliminate this individual which would have an impact on the effects size. The higher starting level of IL6 in this group is not statistically significant different to the other groups. The follow up time points are ok. Therefore, we decided to keep this participant in the study.
- There are some typography and small language errors
We have tried to eliminate these mistakes in the revision
Round 2
Reviewer 1 Report
Comments and Suggestions for Authors/